# Rectal Injury During Radical Prostatectomy: Incidence, Management, and Outcomes in Single-Center Experience

**DOI:** 10.3390/cancers17071129

**Published:** 2025-03-27

**Authors:** Anil Erdik, Haci Ibrahim Cimen, Deniz Gul, Yavuz Tarik Atik, Yasir Muhammed Akca, Fikret Halis, Osman Kose, Hasan Salih Saglam

**Affiliations:** 1Department of Urology, Sakarya Karasu State Hospital, 54500 Sakarya, Turkey; 2Department of Urology, School of Medicine, Sakarya University, 54100 Sakarya, Turkey; hicimen@sakarya.edu.tr (H.I.C.); denizg@sakarya.edu.tr (D.G.); yakca@sakarya.edu.tr (Y.M.A.); halisf@yahoo.com.tr (F.H.); osmankose@sakarya.edu.tr (O.K.); ssaglam@sakarya.edu.tr (H.S.S.); 3Department of Urology, Medstar Antalya Hospital, 07050 Antalya, Turkey; yavuztarikatik@gmail.com

**Keywords:** patient outcome assessment, prostate cancer, rectal wall repair

## Abstract

In this study, we aim to address a critical gap in the literature by analyzing the epidemiology, risk factors, and management strategies for rectal injuries (RIs) during radical prostatectomy (RP). Despite the significant morbidity associated with RIs, limited data exist on protective factors and optimal treatment protocols. We evaluated the relationship between the body mass index (BMI) and RI risk, alongside the impacts of prior pelvic surgeries, tumor aggressiveness, and surgical techniques. Our retrospective cohort analysis revealed that higher BMI significantly reduced RI risk, potentially due to the mechanical barrier effect of perirectal adipose tissue during dissection. Additionally, early intraoperative diagnosis and standardized repair protocols markedly decreased long-term complications such as rectourethral fistula. These findings support the integration of BMI-based risk stratification into clinical practice and the refinement of surgical training programs to address anatomical challenges in high-risk zones. The results of this research may enhance the safety of RP and influence surgical protocols globally by contributing to the development of evidence-based guidelines. Future studies should prospectively validate these associations and histologically examine the adipose tissue barrier hypothesis to optimize preventive strategies.

## 1. Introduction

In European countries and other developed nations, prostate cancer (PCa) ranks as the most common cancer diagnosed in male populations. A study spanning two decades of clinical practice and including 7000 men who underwent prostate biopsy due to suspected prostate cancer (PCa) found that the incidence rate of PCa was 37.7% [1]. Radical prostatectomy (RP) is a standard surgical treatment for localized PCa, which involves the complete removal of the prostate gland. Despite its effectiveness in treating PCa, RP is associated with several potential complications, one of which is rectal injury (RI), a rare but serious complication occurring in 0.5% of RP cases [2], which can significantly affect postoperative outcomes if not promptly identified and managed.

RI can occur primarily during apical dissection, where a posterior plane develops between the rectum and Denonvilliers’ fascia. During radical retropubic prostatectomy (RRP), the critical step that poses a risk is the release of the posterior prostate side from the rectum [3]. Conversely, in laparoscopic (LRP) or robot-assisted radical prostatectomy (RARP), RI is more frequently encountered during the seminal vesicle isolation and dissection of the posterior plane between the prostate and the rectal wall. These critical steps require careful surgical technique to avoid inadvertent trauma to the rectal wall [4].

RI may be identified during surgery, recognized in the postoperative period, or, in some cases, missed intra and perioperatively and present later with infection or rectourethral fistula (RUF). Postoperative diagnosis of rectal injury often presents significant diagnostic challenges. While peritonitis, fever, and abdominal pain or distension are the most frequently observed symptoms, other clinical manifestations, such as ileus, gastrointestinal bleeding, fecal incontinence, passage of enteral contents through the skin, urethra, or vagina, or the presence of urine in the rectum, may also occur, as well as systemic signs including tachycardia, hypotension, leukocytosis, and leukopenia [5]. The occurrence of RI during RP can lead to short- and long-term complications, including bleeding, infection, anastomotic strictures, RUF, fecal incontinence, and sexual dysfunction [5,6,7,8,9]. Delayed recognition of such injuries increases the risks of morbidity and mortality [10].

Recent advances in surgical techniques, particularly the use of robotic systems, have contributed to a reduction in the frequency of RI [2]. However, continuous evaluation and refinement of surgical methods remain necessary to further prevent these injuries.

We hypothesized that the close anatomical proximity of the bladder and prostate to the rectum increases the risk of RI during apical dissection, particularly in the RRP procedures we performed. The primary objective of this study was to examine the clinical data of patients who experienced iatrogenic RI as a complication during RP in our clinic; we aimed to assess the long-term impact of RI on these patients as a secondary objective.

## 2. Materials and Methods

### 2.1. Patient Cohort

This single-center, retrospective study was approved by the Ethics Committee of Sakarya University (institutional review board number: 43012747/050.04/438092-01). All included patients undergoing radical retropubic prostatectomy provided written informed consent for surgery. All procedures performed in studies involving human participants were in accordance with the ethical standards of the institutional and/or national research committee, as well as with the 1964 Helsinki Declaration and its later amendments or comparable ethical standards.

We included a total of 382 patients who underwent RP between October 2012 and October 2024, with each surgery performed by any of six experienced urologic surgeons throughout the study period. Of these, 93 patients (24.3%) underwent RARP, while the remaining 289 patients (75.7%) underwent RRP. All patients were treated for localized PCa, and their perioperative outcomes were meticulously documented.

Patients were included in this study based on the availability of complete medical records and postoperative follow-up data. Data for each patient regarding age, body mass index (BMI), digital rectal examination (DRE) levels and findings, preoperative prostate-specific antigen (PSA) level, prostate volume, percentage of tumors in positive cores, pre-op biopsy Gleason score (GS), number of involved cores, clinical tumor (cT) stage, D’Amico risk classification, pathological tumor stage (pT), lymph node metastasis (LNM), surgical margin status, and comorbidities were also collected to assess potential risk factors for RI.

Postoperative continence status at 12 months was evaluated using daily pad count, and patients were classified as continent (0—safety pad), mildly incontinent (1–2 pad(s)), or severely incontinent (>2 pads). The International Index of Erectile Function (IIEF-5) was used to evaluate patients’ erectile function before surgery and at 1, 3, 6, and 12 months postoperatively [11].

### 2.2. Preoperative Management

Preoperative protocol was conducted according to our national antibiotic prophylaxis guidelines. All patients received a single intravenous 1g dose of cefazoline; if a penicillin allergy had been documented, patients received either clindamycin or vancomycin. Patients were given a clear liquid diet one day before surgery and were instructed to have nothing orally after midnight.

### 2.3. Rectal Injury Repair Technique

All cases of RI were identified intraoperatively through direct visualization of a clear rectal wall defect. In our clinical practice, when RI is suspected, we carefully visualize the dissection area and perform a simultaneous intraoperative DRE. Additionally, following pelvic irrigation with fluid, RI can be more effectively detected by pumping air into the rectum using a Foley catheter. Once an RI was confirmed, the RP procedure was not completed, as rectal repair was performed after specimen retrieval and before vesicourethral anastomosis. The operative field was thoroughly irrigated with an antibiotic solution to minimize the risk of infection.

Repair of the rectal defect was performed in two layers (inner mucosa and outer seromuscular layer) using 2-0 Vicryl sutures. After the repair was completed, the operative site was inspected for potential leaks by insufflating air through a rectal catheter while the pelvis was filled with sterile saline [12].

Following the rectal repair, vesicourethral anastomosis was performed by a urologic surgeon. For open prostatectomy, a 4-0 Monocryl suture was used, whereas a 2-0 V-Lock suture was employed for robot-assisted radical prostatectomy utilizing the Van Velthoven technique [13]. The integrity of the anastomosis was tested by filling the bladder with 250 mL of sterile saline.

A Jackson–Pratt drain was routinely placed in the space of Retzius to monitor and manage postoperative fluid accumulation. Routine anal dilation was not performed as part of the surgical protocol. In cases where an anastomotic leak was observed, additional reinforcing sutures were placed, or the anastomosis was repeated to ensure a watertight closure.

### 2.4. Postoperative Care

Our routine postoperative pathway for non-RI patients consisted of antibiotics perioperatively for 24 h, a clear liquid diet on the first post-op day, and early ambulation. On the other hand, patients with an RI were switched to third-generation cephalosporin and metronidazole and generally started on a clear liquid diet on postoperative day 2, with the diet advanced as tolerated. The Jackson–Pratt drain was removed when output was minimal. Cystography was routinely performed on postoperative day 21 in patients with RI to confirm or exclude RUF. In the absence of contrast leakage, the catheter was subsequently removed. Readmission within 30 days, mortality, and complications up to 90 days postoperatively were assessed using the Clavien–Dindo (CD) system, where grades 3–5 denote major complications and grades 0–2 indicate minor or no complications. Postoperative follow-up protocols necessitate vigilant monitoring for signs of RI, including, but not limited to, abdominal pain, hypotension, fever, tachycardia, peritonitis, leukocytosis, or leukopenia. A high clinical suspicion of RI should be maintained in cases of enteric fluid leakage through cutaneous or urethral routes, involuntary fecal discharge, or pneumaturia. Immediate intervention is critical upon identification of these findings to mitigate complications.

### 2.5. Statistical Analyses

Comparisons between the patients in the RI and non-RI groups were conducted using the Mann–Whitney U test for continuous variables and the chi-square test for categorical variables. Multivariate linear and logistic regression analyses were employed to predict RI risk. All features identified as statistically significant predictors for RI through univariate analysis were included in the final multivariate models. Statistical analyses were conducted using IBM SPSS Statistics for Windows (version 26.0, IBM Corp., Armonk, NY, USA). All tests were two-sided, with a *p*-value of <0.05 considered statistically significant.

## 3. Results

Out of the 382 patients who underwent RP, RI was identified in 9 cases, resulting in an incidence rate of 2.4%. Notably, all RIs occurred in patients who underwent open procedures, while no RIs were observed in the RARP group. The mean duration from transrectal biopsy to operation was 62.4 ± 30.2 days for the RI group. The preoperative GSs were 6, 7, and 9 in six, two, and one patient(s), respectively. None of the nine patients with an RI had undergone previous prostatic surgery, preoperative radiotherapy (RT), or androgen deprivation therapy (ADT). The characteristics of patients who experienced RI are summarized in Table 1.

### 3.1. Postoperative Characteristics of the Patients

Age; preoperative PSA, PV, DRE, and GS; D’Amico risk classification; and presence of ED were similar in both groups (*p* > 0.05). On the other hand, patients with an RI had a lower BMI. Table 2 shows the preoperative characteristics of the patients according to their RI status.

### 3.2. Surgical and Postoperative Outcomes Following Rectal Injury Repair

All RIs were identified and primarily repaired intraoperatively by a general surgeon. The mean length of rectal injuries was 2.0 ± 1.0 cm, with four injuries (44.4%) located in the anterior region of the rectum. The other RIs were two cases during apical prostate dissection, one injury to the anterolateral rectal wall, one injury to the lateral rectal wall, and one injury involving the rectourethral muscles. Among RI patients, the catheterization time was uniformly set at 21 days, and the mean hospital stay was 7.6 ± 2.2 days. Among RI patients, one patient (11.1%) developed an RUF on the 8th day, which necessitated a colostomy. No patients who experienced RI required hospital readmission within the 30-day postoperative period due to related RP complications.

### 3.3. Pathological Results

Among patients with RIs, the mean weight of the surgical specimens (i.e., prostate and seminal vesicles) was 49.0 ± 10.3 g. The pathological T stage distribution included four patients (44.4%) with pT2c disease and five patients (55.5%) with pT3b disease. The mean GS was 6.8 ± 1.2. Pathological findings also included LNM in two patients (22.2%), capsule invasion in four patients (44.4%), and positive surgical margins (PSM) in five patients (55.5%). Notably, in the patient who developed an RUF postoperatively, all of the aforementioned criteria—LNM, capsule invasion, and PSM—were present.

### 3.4. Predictive Factors for Rectal Injury Following Radical Prostatectomy

Among the variables identified as predictive factors for RI in the univariate analysis, the BMI remained the most predictive factor in the multivariate analysis (Table 3).

### 3.5. Long-Term Patient Outcomes After Rectal Injury

No major complications were observed within 30 days following rectal injury repair; however, one patient (11.1%) developed a minor complication (Clavien–Dindo grade 2) within the 90-day postoperative period.

During further follow-up, one of the nine patients (11.1%) with a documented intraoperative RI following non-salvage RP underwent a diverting colostomy six months after RI repair. The colostomy was reversed after diagnostic confirmation of fistula closure. The patient had a pathological tumor stage of T3b, with LNM and PSM. Consequently, the patient was referred for RT three months after colostomy reversal. Unfortunately, an RUF developed in the third month following RT. Among RI patients, one had mild incontinence (11.1%), and one had severe incontinence (11.1%), but no significant difference was found between the groups 12 months after surgery (Table 4); on the other hand, IIEF-5 scores were lower at baseline (Figure 1), and there was a significant difference at 12 months postoperatively (Table 5).

## 4. Discussion

RI is a known but uncommon complication of RP; its incidence has been widely reported in the literature, with variability based on surgical technique and expertise. A recent meta-analysis determined the overall incidence of RI to be 0.58%, with the highest rates observed in RRP and LRP cases, with fewer occurring in RARP [4]; RI incidence was also significantly higher in low-volume centers performed by low-volume surgeons [4]. The 2024 European Association of Urology guidelines reported organ injury rates during RP as 0.4% for RARP, 2.9% for LRP, and 0.8% for RRP [14]. Recent studies have highlighted varying incidence rates of RI depending on the surgical approach for RP; for example, population-based analyses have reported RI rates of 0.6% for RRP, 0.4% for LRP, and 0.3% for RARP [2]; similarly, another study confirmed these findings, demonstrating higher RI rates in RRP (up to 0.55%) compared to RARP (0.27%) [15].

Surgeon experience and learning curve are significant RI risk factors, primarily in RRP and LRP [16]. Heinzer et al. evaluated open RP and revealed a 7.8% RI rate among patients operated on during the initial phase of the surgeon’s learning curve, compared to 2% in patients operated on later [17]; in contrast, Kheterpal et al. found no link between RI incidence and surgeon experience in over 4000 RARP cases [18]. Hospital and surgeon caseload volumes have been identified as significant factors influencing RI rates during RP [16], with Barashi et al. demonstrating that institutions performing a high annual volume of RP (>43 cases/year) exhibit a significantly lower risk of RI compared to low-volume centers (1–43 cases/year) [2]; similarly, Van den Broeck et al. reported that an annual caseload exceeding 86 procedures is associated with reduced complication rates [19]. In a related analysis, Schmitges et al. highlighted the impact of individual surgeon experience, revealing that low-volume surgeons (<7 cases/year) carry a higher RI risk relative to very high volume surgeons (51 cases/year) [20]. Consistent with these findings, all RIs in our study occurred during RRP procedures, with no cases observed in RARP procedures. Our cohort exhibited a rectal injury incidence of 2.4%, which, although higher than the meta-analytic average, highlights the inherent technical challenges of open surgical approaches in PCa management. Furthermore, the relatively low caseloads of both our institution and the multiple surgeons involved in PCa treatment may have contributed to the elevated RI incidence observed.

Previous benign prostatic hyperplasia (BPH) surgery and large PV have been associated with an increased risk of RI [2]; however, a recent meta-analysis found no significant relationship between a history of BPH surgery and the risk of RI [4]. In our cohort, ten patients had a history of BPH surgery (2.6%), yet none of the patients who developed RI had undergone prior BPH surgery, consistent with the existing literature.

Risk factors for RI in PCa management extend beyond surgical technique. Post-primary tumor treatments, particularly RT, are strongly associated with RI risk. In the context of PCa, local therapies such as brachytherapy, high-intensity focused ultrasound, and cryotherapy, alongside systemic treatments including ADT, have been implicated in increased RI incidence [16]; additionally, prior pelvic fractures, rectal surgeries, or chronic pelvic inflammation may contribute to periprostatic fibrosis, a fibrotic remodeling that can obscure anatomical planes, particularly the posterior surgical dissection zone, thereby elevating intraoperative RI risk. Collectively, these multifactorial risks underscore the complexity of prostate surgery, especially in patients with a history of pelvic interventions or multimodal PCa therapies [21]. Patients with a history of inflammatory bowel disease or prior abdominal surgeries, such as colorectal or pelvic procedures, may be at higher risk of RI during RP. While minimally invasive approaches can be considered, some authors advocate for direct retropubic techniques in patients with extensive or complex surgical histories in order to mitigate the risk of complications, including intestinal injury [22]. In contrast to the established risk factors highlighted in prior studies, none of the patients in our cohort had undergone preoperative RT or ADT prior to RP. Notably, only one patient (11.1%) among those who sustained RIs had a history of abdominal surgery—specifically, a prior intervention for ileus management, a finding which suggests that, while systemic or local therapies (e.g., radiation, ADT) are well-documented contributors to periprostatic fibrosis and RI risk, isolated surgical interventions such as ileus-related abdominal procedures may also play a context-dependent role in select cases.

Patient-specific factors, such as obesity and advanced age, have been inconsistently linked to RI in RP, though robust clinical evidence remains limited [5]. While some studies associate obesity with poorer oncologic outcomes and perioperative complications post-RP [23,24], others paradoxically suggest a protective effect against metastasis and surgical morbidity, termed the metabolic paradox of obesity [25,26]. For instance, Barashi et al. identified African ancestry and BPH as independent risk factors for RI, whereas obesity, high-volume surgical centers, and robotic-assisted approaches were correlated with reduced RI rates [2]. The authors hypothesize that mild-to-moderate obesity may enhance metabolic resilience or immune modulation, while perirectal adipose tissue could mechanically buffer rectal–prostatic interfaces during dissection [2]. In our cohort, each unit increase in the BMI reduced RI risk 1.45-fold (OR: 0.68, 95% CI: 0.47–0.99), aligning with the proposed protective role of adiposity in this context. Our findings underscore the need for risk stratification models that integrate the BMI and periprostatic anatomy, particularly in patients with prior pelvic interventions or metabolic comorbidities; however, while adiposity may offer mechanical advantages during RP, its long-term metabolic consequences (e.g., insulin resistance, hypertension, dyslipidemia, atherosclerotic cardiovascular diseases, and type II diabetes mellitus) necessitate a balanced surgical approach. We advocate for prospective studies to validate these associations and refine patient selection criteria for open versus robotic techniques based on body habitus.

A transrectal prostate biopsy performed shortly before RP may distort the rectal–prostatic interface due to residual inflammation or surgical adhesions. Yildirim et al. proposed that delaying surgery for a defined period post-biopsy facilitates anatomical plane dissection and reduces the risk of RI [27]. The current literature hypothesizes that a minimum interval of one month allows for the resolution of biopsy-induced tissue inflammation and advises minimizing RI risk preoperatively. In our cohort, the mean interval from transrectal biopsy to surgery in the RI group was 62.4 ± 30.2 days, aligning with existing evidence supporting extended intervals (>4 weeks) for optimal tissue healing. These findings reinforce the importance of adhering to recommended waiting periods between biopsy and RP.

RARP offers technological advantages that enhance surgical precision and safety. Features such as three-dimensional visualization, optical magnification, the flexibility of 0° and 30° lens angles, and the EndoWrist technology allow surgeons to operate on the seminal vesicles, the posterior prostate surface, and the apex level under direct vision. These capabilities significantly reduce the risk of RI [4]. Comparative studies consistently demonstrate the superiority of RARP over open approaches, with significant advantages in operative efficiency (e.g., shorter surgical duration), reduced intraoperative blood loss, and lower rates of perioperative morbidity, including shorter hospital stays and fewer postoperative complications [28], benefits which are particularly critical in contemporary practice, where prostatectomies are increasingly performed on patients with advanced-stage or high-risk prostate cancer, necessitating precision and minimized surgical trauma [29]. In our cohort, five of the 93 RARP patients had a history of BPH surgery, and another five had undergone prior rectal surgeries; nevertheless, none of these patients experienced RI or required conversion to open surgery.

Considering pathological variables, RI appears to be more frequent in patients with locally advanced disease, high GS, LNM, and PSM [2,15]. These features may reflect the challenging nature of surgical dissection in cases of extensive disease, where distorted anatomy and periprostatic fibrosis and/or adhesion may contribute to an increased risk of RI [5]. Our findings reveal that a significant proportion of RI patients exhibited high-risk pathological features, including 44.4% with the T3b disease, 55.5% with PSM, 22.2% with LNM, and 44.4% with capsule invasion. Interestingly, none of the pathological variables in our cohort directly correlated with RI occurrence, which suggests that, while adverse pathological features may exacerbate postoperative complications, their presence does not inherently predispose a patient to RI during surgery; notably, the patient who developed a RUF postoperatively demonstrated all four risk factors.

Prospective randomized trials comparing primary repair to fecal diversion (colostomy) in RIs remain absent, and standardized decision-making algorithms are yet to be established [16]; however, a systematic review by Leevan et al., synthesizing existing evidence, found no significant disparity in clinical outcomes between primary repair and diversion strategies [5]. Drawing parallels from trauma surgery protocols, the intraoperative management of RIs identified intraoperatively typically favors primary suturing of the defect; while current evidence supports the feasibility of primary repair for select RIs, the absence of comparative studies necessitates cautious individualization of treatment. Prospective studies stratifying patients by injury severity and surgical context are warranted to refine these guidelines. The management and prognosis of RI following RP vary significantly based on the timing of their detection, either intraoperatively or postoperatively; early detection of RI, whether intraoperatively or in the early postoperative period, has been strongly associated with a substantial reduction in severe postoperative complications and the subsequent development of an RUF [30]. Patients diagnosed postoperatively experienced a significantly higher burden of surgical interventions, prolonged hospitalization, and elevated overall mortality rates compared to those whose conditions were identified intraoperatively [5]. In our cohort, 10% of patients diagnosed with RI underwent fecal diversion, a rate aligning with prior reports in the literature.

The timely detection of RI during surgery is paramount to minimizing postoperative complications and ensuring optimal patient outcomes [4]. Intraoperative measures to identify RI might be routinely implemented, with one practical approach involving placing a rectal probe or sponge stick at the beginning of the procedure to assist in visualizing any rectal mucosal perforations [4]. If RI is suspected or if a thin rectal wall is observed, a pneumatic test is essential. Initially, the pelvis should be thoroughly irrigated with normal saline to clear blood clots and expose any actively bleeding vessels. Following this, the pelvis can be filled with saline, and air insufflation into the rectum, facilitated with a rectal probe, can reveal the presence of a rectal defect through escaping air bubbles; in the absence of a rectal probe, a rectal Foley catheter may be used for air insufflation [31,32]. DRE further helps to identify the extent of RI; using a two-finger technique, surgeons can thoroughly evaluate the defect, ensuring precise localization and improving the accuracy of defect closure [33]. These awareness-enhancing measures assist surgeons in managing RI more effectively during surgical procedures. Colostomy diversion should not be regarded as the standard of care for all patients with intraoperatively detected RI; however, it should be strongly considered in cases involving larger rectal defects (>2 cm), prior pelvic RT, previous BPH-related surgeries, or suspected rectal infiltration by PCa [4].

Postoperative management may include initiating a clear liquid diet on the first postoperative day, with a gradual transition to a regular diet by the third day. Broad-spectrum antibiotic therapy, such as penicillin or cefuroxime combined with metronidazole, is highly recommended to minimize the risk of infection [15]. While conventional postoperative protocols recommend initiating oral liquid intake as early as postoperative day 1, our clinical practice adopts a more cautious approach for patients with RI, delaying clear liquid diets until postoperative day 2 to mitigate risks of early enteral stimulation on healing tissues. In patients with RI, catheterization for at least two weeks should be considered, with catheter removal planned only after confirming the absence of leakage on cystography [15,31,34]. However, despite intraoperative identification and repair of RI, approximately 10% of patients may still develop RUF in the postoperative period [35], and the overall incidence of RUF after RP was <0.1% [15]; otherwise, in our clinic, we standardized the catheterization duration to an extended 21 days for all patients with RI, a decision which aligns with our institutional policy. Furthermore, all RI patients underwent cystography before catheter removal to ensure the integrity of the repair. In one case, fecal discharge was noted from the urethral catheter shortly after the commencement of oral feeding. Contrast-enhanced cystography subsequently revealed colonic passage of the contrast agent, prompting immediate cessation of oral intake. Following multidisciplinary evaluation, a sigmoid loop colostomy was performed on postoperative day 8. Within our cohort, 10% of patients with RI required fecal diversion, a rate consistent with those reported in large-scale surgical studies [2,15]. Notably, while the observed RI incidence in our series exceeds meta-analytic benchmarks, the postoperative RUF rate remained comparable to those in the established literature, suggesting comparable efficacy in managing severe complications despite higher baseline injury rates.

The literature on functional outcomes following RI remains sparse, yet the association between rectal cancer treatment and sexual dysfunction is well documented. Proposed mechanisms include intraoperative nerve traction or ischemia due to surgical techniques that compromise pelvic vascular supply [36,37], as well as neurovascular toxicity from adjuvant therapies, such as chemotherapy and radiation [38]. Furthermore, the pathophysiology of the primary disease (e.g., tumor-induced inflammation) may independently exacerbate sexual morbidity. Recent systematic reviews estimate that approximately 35% of patients develop moderate-to-severe erectile dysfunction within the first postoperative year after rectal cancer surgery, with minimal improvement over time [39]. In our cohort, patients with RI exhibited notably low baseline IIEF-5 scores (median: 8), which persisted at the 12-month follow-up (median: 7), underscoring the chronicity of sexual dysfunction in this population.

## 5. Study Limitations

Our study was conducted in a single center, which may limit the generalizability of the results to other settings or patient populations. Due to the retrospective design, some important data may not have been fully assessed or may be incomplete. The RRP surgeries were performed by different surgeons, which could have introduced heterogeneity in surgical techniques, potentially affecting the incidence of RI. Finally, long-term outcomes and quality of life assessments were not evaluated in our study, limiting the understanding of the full impacts of complications such as rectourethral fistula.

## 6. Conclusions

The results of this study offer critical insights into the incidence, risk factors, and management strategies for RI during RP. Our findings underscore that RI, while a concerning iatrogenic complication, can achieve favorable long-term outcomes when promptly identified intraoperatively and managed through standardized surgical protocols. A notable contribution of our research is the identification of the BMI as a protective factor against RI, which is potentially attributable to the mechanical buffering effect of perirectal adipose tissue during dissection. While our reported RI incidence diverges slightly from the existing literature, this discrepancy may reflect differences in patient demographics, surgical procedures, and institutional volume. Crucially, our results reinforce the necessity of early detection through vigilant intraoperative practices, including rectal mucosal surveillance and pneumatic testing, which are pivotal in mitigating severe postoperative sequelae such as RUF. Our findings support the integration of preoperative risk stratification models—incorporating the BMI, surgical history, and tumor characteristics—to optimize patient selection and procedural planning. Future research should focus on multicenter collaborations to validate these associations and explore novel preventive strategies.

## Figures and Tables

**Figure 1 cancers-17-01129-f001:**
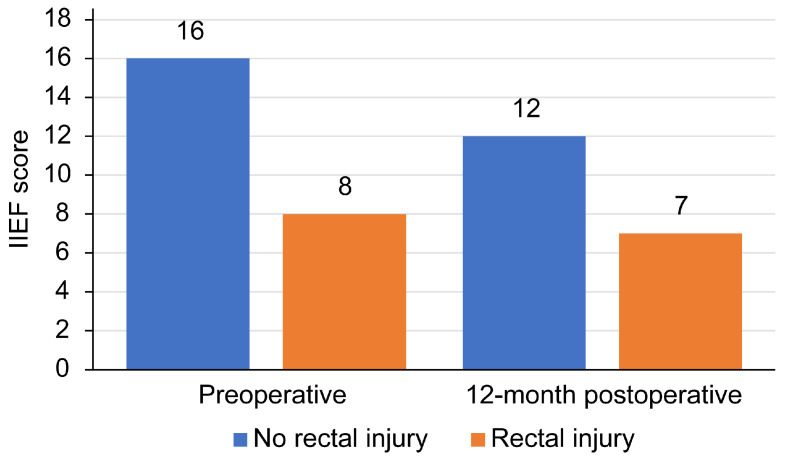
Comparison of IEFF scores preoperatively and at 12-month postoperative follow-up.

**Table 1 cancers-17-01129-t001:** Clinical characteristics of patients with rectal injury during radical prostatectomy.

ID	A	B	C	D	E	F	G	H	I	J	K	L	M	N
1	57	Nodule on both sides	100	50	40	3 + 3	T3b	Positive	4	6	Yes	No	No	2
2	72	Nodule on the right side	95	17	58	4 + 5	T3b	Positive	2	9	No	No	No	3
3	73	Normal	67	4.3	55	4 + 3	T2c	Negative	2	8	No	No	No	2
4	60	Nodule on the right side	25	6.5	30	3 + 3	T2c	Negative	1	7	No	No	No	2
5	69	Nodule on the right side	33	8.1	30	3 + 3	T2c	Positive	1	6	No	No	No	2
6	60	Normal	33	7	41	3 + 3	T2c	Negative	2	7	No	No	No	2
7	60	Normal	25	7	61	3 + 3	T2b	Negative	2	6	No	No	No	2
8	53	Nodule on both sides	92	9	48	4 + 5	T3b	Positive	3	13	No	*	Yes	9
9	60	Nodule on both sides	92	69	60	3 + 3	T3b	Positive	1	7	No	No	No	2

A: age at time of operation; B: digital rectal examination findings; C: positive core (%); D: preoperative PSA (ng/mL); E: prostate volume; F: Pre-op biopsy Gleason score; G: pathological T stage; H: surgical margin status; I: rectal defect (cm); J: hospitalization (days); K: prior abdominal surgery; L: postoperative complications; * colostomy performed on the 8th postoperative day—all patients underwent primary repair; M: rectourethral fistula; N: oral diet (postoperative days).

**Table 2 cancers-17-01129-t002:** Comparison of demographic characteristics and preoperative data.

Variable	No Rectal Injury (n = 373)	Rectal Injury (n = 9)	*p*-Value
Age (years) *	65 (8)	61 (12)	0.329 ^a^
BMI (kg/m^2^) *	27.42 (2.84)	26.05 (0.68)	0.019 ^a^
Preoperative PSA level (ng/mL) *	8.2 (8.3)	8.1 (26.7)	0.552 ^a^
Prostate volume (mL) *	50 (27)	49 (17)	0.746 ^a^
DRE, grade ^#^			0.895 ^b^
1	182 (48.8)	5 (55.6)	
2	153 (41.0)	3 (33.3)	
3	38 (10.2)	1 (11.1)	
Preoperative GS ^#^			0.376 ^b^
≤6	258 (69.2)	6 (66.7)	
7–8	81 (21.7)	1 (11.1)	
≥9	34 (9.1)	2 (22.2)	
D’Amico risk classification ^#^			0.368 ^b^
Low	179 (48.0)	3 (33.3)	
Intermediate	107 (28.7)	2 (22.2)	
High	87 (23.3)	4 (44.4)	
Presence of preoperative ED ^#^	257 (68.9)	9 (100)	0.062 ^b^

* Median (IQR), ^#^ n (%), ^a^ Mann–Whitney U test, ^b^ chi-square test. BMI: body mass index; DRE: digital rectal examination; GS: Gleason score; PSA: prostate-specific antigen.

**Table 3 cancers-17-01129-t003:** Univariate and multivariate analyses of risk factors for rectal injury during radical prostatectomy.

Variable	Univariate Analysis		Multivariate Analysis	
	OR (95% CI)	*p*-Value	OR (95% CI)	*p*-Value
Age	0.954 (0.859–1.060)	0.379	0.950 (0.850–1.061)	0.361
BMI	0.675 (0.471–0.969)	0.033	0.689 (0.478–0.995)	0.047
DRE				
(Reference: Grade 1)				
Grade 2	0.714 (0.168–3.035)	0.648	0.656 (0.137–3.140)	0.598
Grade 3	0.958 (0.109–8.434)	0.969	1.044 (0.070–15.640)	0.975
Prostate Volume	0.988 (0.956–1.020)	0.452	0.989 (0.949–1.030)	0.586
Preoperative PSA	1.020 (0.992–1.049)	0.172	1.014 (0.978–1.052)	0.451
Preoperative GS				
(Reference: ≤6)				
7	0.531 (0.063–4.475)	0.560	0.425 (0.044–4.105)	0.460
≥8	2.529 (0.491–13.037)	0.267	2.180 (0.271–17.559)	0.464
D’Amico risk classification				
(Reference: Low risk)				
Intermediate risk				
High risk	1.115 (0.183–6.782)	0.906	1.641 (0.244–11.054)	0.611
	2.473 (0.601–12.527)	0.193	1.833 (0.242–13.859)	0.557

BMI: body mass index; DRE: digital rectal examination; GS: Gleason score; PSA: prostate-specific antigen.

**Table 4 cancers-17-01129-t004:** Analysis of postoperative continence status.

Continence Status	No Rectal Injury (n = 373)	Rectal Injury (n = 9)	*p*-Value
12 months *			
Continent	246 (71.5)	7 (77.8)	0.870 ^a^
Mildly incontinent	67 (19.5)	1 (11.1)	
Severely incontinent	31 (9.0)	1 (11.1)	

* n (%), ^a^ Pearson Chi-Square test.

**Table 5 cancers-17-01129-t005:** Comparison of IEFF scores preoperatively and 12 months after radical prostatectomy.

	No Rectal Injury (n = 373)	Rectal Injury (n = 9)	*p*-Value
Preoperative IIEF-5 *	16 (10)	8 (6)	<0.001 ^a^
12-month Postoperative IIEF-5 *	12 (11)	7 (4)	

* Median (IQR); ^a^ Wicoxon test.

## Data Availability

The datasets generated during and/or analyzed during the current study are available from the corresponding author upon reasonable request.

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
