# Peer review of "Rectal Injury During Radical Prostatectomy: Incidence, Management, and Outcomes in Single-Center Experience"

_cancers, 2025, doi:10.3390/cancers17071129_

Round 1
Reviewer 1 Report
Comments and Suggestions for Authors
In the introduction the authors should:
-say something about the epidemiology (https://doi.org/10.21873/anticanres.13254)
-elaborate the severity and implications of the complications discussing also the potential long-term consequences of RI, such as fecal incontinence, anastomotic strictures, and sexual dysfunction.
-state the research question, the aims and objectives.
In the discussion the authors should:
- compare the study's findings with those of other studies on RI during RP
-discuss more in detail the clinical implications of the findings
The authors mention statistical analysis but do not provide specific details on the methods used; they should specify the statistical tests used to analyze the data, including any adjustments made for multiple comparisons.
The authors don't report potential risk factors, such as age, BMI, PSA level, prostate volume, or Gleason score. It would be helpful to analyze these factors to identify any potential associations with RI.
The authors report on the short-term outcomes but they don't provide any information on long-term outcomes (urinary incontinence, erectile dysfunction, or quality of life). It would be helpful to follow up on these patients to assess the long-term impact of RI.
Comments on the Quality of English LanguageThe English in the manuscript is generally well written however pay attention to some sentences are quite long and complex and Tto a few minor punctuation errors
Author Response
Comments and Suggestions for Authors
In the introduction the authors should:
Comment 1: say something about the epidemiology (https://doi.org/10.21873/anticanres.13254)
Response 1: Thank you for pointing out. We agree with this comment. The revised manuscript this paragraph can be found introduction section- page number 2 and line 46-49 (marked as highlighted )
Comment 2: elaborate the severity and implications of the complications discussing also the potential long-term consequences of RI, such as fecal incontinence, anastomotic strictures, and sexual dysfunction.
Response 2: We agree with this comment. Additionally, the potential long-term consequences of RI are also mentioned in the introduction section. Page number 2 and line 63-72 (marked as highlighted )
Comment 3: state the research question, the aims and objectives.
Response 3: Thank you for pointing out. We have modified paragraph to emphasize this point. Page number 2 and line 76-80 (marked as highlighted )
In the discussion the authors should:
Comment 4: compare the study's findings with those of other studies on RI during RP
Response 4: Thank you for pointing out. Our study’s findings was compared to other studies in discussion section. Page number 8 and line 249-267, 270-272, page number 9 and line 295-314, page number 11 and line 400-409, 410-421 (marked as highlighted )
Comment 5: discuss more in detail the clinical implications of the findings
Response 5: Thank you for pointing out. Our study’s findings discussed more in detail in the discussion section. Page number 8 and line 249-267, 270-272, page number 9 and line 287-294, 295-314, 315-324, page number 10 and line 329-336, 350-358, page number 11 and line 400-409, 410-421 (marked as highlighted )
Comment 6: The authors mention statistical analysis but do not provide specific details on the methods used; they should specify the statistical tests used to analyze the data, including any adjustments made for multiple comparisons.
Response 6: We agree with this comment. The statictical analyses was added to methods section. Page number 4 and line 153-159 (marked as highlighted )
Comment 7: The authors don't report potential risk factors, such as age, BMI, PSA level, prostate volume, or Gleason score. It would be helpful to analyze these factors to identify any potential associations with RI.
Response 7: We agree with this comment. The predictive factors for rectal injury following radical prostatectomy was reported and added to results section. Page number 6 and line 204-207 (marked as highlighted )
Comment 8: The authors report on the short-term outcomes but they don't provide any information on long-term outcomes (urinary incontinence, erectile dysfunction, or quality of life). It would be helpful to follow up on these patients to assess the long-term impact of RI.
Response 8: We agree this comment. The long-term outcomes was reported and added to results section. Page number 7 and line 212-225 (marked as highlighted )
Comments on the Quality of English Language
The English in the manuscript is generally well written however pay attention to some sentences are quite long and complex and Tto a few minor punctuation errors
Reviewer 2 Report
Comments and Suggestions for Authors
This an interesting yet not novel paper on a truly urological nightmare, i.e. rectal injury that occurs at the time of radical prostatectomy (RP).
Although it is rare complication in its nature, it is not only connected with prolonged hospital stay, but also may end up with additional surgical interventions, not to mentioned thorough impact on patients’ quality of life.
Here, the authors presented a series of 382 patients analyzed retrospectively that underwent RP (mainly RRP, over 75%). Please see my comments that were listed below:
It is hard to identify and localize all the rectal wall defects based on pure macroscopic assessment. Did you use additional techniques, e.g. DRE or pumping air via rectal canula?
Can you characterize the defects you noticed further (e.g. length, side, etc.)?
Were there any cases that needed colostomy? If so, based on which grounds have you decided about that?
A recent meta analysis by Romito et al (10.1016/j.euros.2023.03.017) noted that the incidence was higher in those who underwent open/laparoscopic approach. In your study, all the injuries were found in case of RRP. Please further comment on that (it is relatively briefly mentioned in page 4, line 145). Is it the advanced stage or inflammation or recent transrectal biopsy that may increase the risk? What differs RRP and RARP that even during learning curve for robotic system it is observed in fewer cases? Which robotic system do you use?
What is the follow up of early repair? What are you results as for example fistula formation?
Please correct several typing errors, e.g. protocol.terile saline (page 2, line 90).
Author Response
Comments and Suggestions for Authors
This an interesting yet not novel paper on a truly urological nightmare, i.e. rectal injury that occurs at the time of radical prostatectomy (RP).
Although it is rare complication in its nature, it is not only connected with prolonged hospital stay, but also may end up with additional surgical interventions, not to mentioned thorough impact on patients’ quality of life.
Here, the authors presented a series of 382 patients analyzed retrospectively that underwent RP (mainly RRP, over 75%). Please see my comments that were listed below:
Comment 1: It is hard to identify and localize all the rectal wall defects based on pure macroscopic assessment. Did you use additional techniques, e.g. DRE or pumping air via rectal canula?
Response 1: Thank you for this pointing out. When we suspect rectal injury, we visualize the dissection area with simultaneous intraoperative digital rectal examination. Additionally, after irrigating the pelvic cavity with fluid, we pumping air into the rectum using a rectal Foley catheter to more readily identify rectal injury. The paragraph was added to methods section. Page number 3 and line 115-119 (marked as highlighted )
Comment 2: Can you characterize the defects you noticed further (e.g. length, side, etc.)?
Response 2: The other rectal injury occured included two cases during prostatic apical dissection in four patients, one at the anterolateral rectal wall, one at the lateral rectal wall, and one involving the rectourethral muscles. This paragraph was added to results section. Also, the length of rectal defects are mentioned in Table 1. Page number 5 and line 187-189 (marked as highlighted )
Comment 3: Were there any cases that needed colostomy? If so, based on which grounds have you decided about that?
Response 3: Thank you for this pointing out. In one patient, fecal drainage was observed draining from the urethral tube following the initiation of oral diet. Oral intake was immediately stopped upon detection of contrast passage into the colon during cystography. A general surgery consultation was requested, and a sigmoid loop colostomy was performed on the 8th postoperative day. Page number 5 and line 216-225 (marked as highlighted )
Comment 4: A recent meta analysis by Romito et al (10.1016/j.euros.2023.03.017) noted that the incidence was higher in those who underwent open/laparoscopic approach. In your study, all the injuries were found in case of RRP. Please further comment on that (it is relatively briefly mentioned in page 4, line 145). Is it the advanced stage or inflammation or recent transrectal biopsy that may increase the risk? What differs RRP and RARP that even during learning curve for robotic system it is observed in fewer cases? Which robotic system do you use?
Response 4: As you mentioned all rectal injuries occured in open / RRP surgeries. Page number 8 and line 242-248 (marked as highlighted )
Comment 5: What is the follow up of early repair? What are you results as for example fistula formation?
Response 5: Agree. Follow up contains active monitoring of the patient and suspicion of rectal injury in case of abdominal pain, hypotension, fever, tachycardia, peritonitis, leukocytosis, and/or leukopenia; fast management of injury in case of drainage of enteral contents through the skin, urethra, or fecal incontinence. After patient’s discharge, during further follow-up, one of the nine patients (11.1%) with documented intraoperative RI following non-salvage RP underwent a diverting colostomy six months after RI repair. The colostomy was reversed after diagnostic confirmation of fistula closure. Unfortunately, a RUF developed in the third month following radiotherapy. The long-term patients outcomes after rectal ınjury subtitle was added to methods and results section. Page number 4 and line 143-151, page number 6 and line 192-194 (marked as highlighted )
Comment 6: Please correct several typing errors, e.g. protocol.terile saline (page 2, line 90).
Response 6: Agree. The typing errors was corrected. Page number 3 and line 125 (marked as highlighted )
Reviewer 3 Report
Comments and Suggestions for Authors
Compliments to the authors for presenting a large single center study looking at rectal injury during radical prostatectomy. This is a good study.
I suggest some changes and corrections:
- Line 16: is there any change in vesicourethral anastomosis in patients where rectal injury is recorded?
- Line 20: To say that the mean catheterization time was 21 days may not be correct as catheter was placed for 21 days as per protocol.
- You aimed to assess the risk factors such as prior pelvic surgery and high-risk pathological features; the results of this assessment needs to be mentioned in the result section of the abstract.
- Line 22: Is pneumatic testing a preventive measures or a diagnostic measure?
- Line 43: How would we know about a rectal injury if it remains undetected altogether?
- Line 78: you have written that once the RI was confirmed, the RP procedure was completed. The RP procedure was not completed as the rectal repair was performed after specimen retrieval before vesico-urethral anastomosis.
- Line 91: some typo (surgical protocol.terile saline.).
- Line 108: You have mentioned that none of the patients with a RI had undergone previous prostatic surgery or preoperative radiotherapy or hormonal therapy. How many patients in your series of 382 patients had this history?
- In table 1 (line 114): Spelling of complications is wrong.
- Table 1 (line 114): Change Post-operative 8th day perform colostomy to Colostomy performed on the 8th post-operative day.
- Line 118: Instead of general surgery surgeon, better to say gastro-intestinal surgeon or general surgeon.
- Line 119: you have mentioned that four injuries were in the anterior region of the rectum; where were the other injuries?
- Line 122: change No patients were readmission due to complications to No patients needed readmission due to complications.
- Line 125: what do you mean by mean pathological prostate volume?
- Line 128: LVI means lymphovascular invasion
- Line 136: you have said RI incidence was significantly higher in low-volume centres and in low-volume surgeons. I think you mean in low-volume centres performed by low-volume surgeons.
- Line 196: change this sentence, To identify the extent of RI, using DRE further enhances. I suggest, DRE further helps to identify the extent of RI.
- Your post-op suggestions in Discussion do not match what you have described in your care path way. Line 204 you have mentioned that clear liquid diet can be initiated on the first post- operative day but in the post-op care section (line 98) you suggest to generally start clear liquid diet on postoperative day 2. Please check. Why is cystogram needed in patients with rectal injury (Line 209)? Did you routinely do it? Is it a cystogram or a peri-catheterogram? In Discussion you suggest catheter for two weeks but you have kept for three weeks.
- In Abbreviations: Is DRE Digital rectal examination levels?
Mentioned in the comments to Authors
Author Response
Comments and Suggestions for Authors
Compliments to the authors for presenting a large single center study looking at rectal injury during radical prostatectomy. This is a good study.
I suggest some changes and corrections:
- Line 16: is there any change in vesicourethral anastomosis in patients where rectal injury is recorded?
Respond 1: No. We observed any change in vesicourethral anastomosis in patients with recorded rectal injury.
- Line 20: To say that the mean catheterization time was 21 days may not be correct as catheter was placed for 21 days as per protocol
Respond 2: We agree and we removed this sentence into the abstract.
- You aimed to assess the risk factors such as prior pelvic surgery and high-risk pathological features; the results of this assessment needs to be mentioned in the result section of the abstract.
Respond 3: We agree. Risk factors was added to abstract. Page number 1, line number 31-37 (marked as highlighted)
- Line 22: Is pneumatic testing a preventive measures or a diagnostic measure?
Respond 4: Agree. Preventive measures revised to a diagnostic measues. It was revised. Page number 1, line number 40 (marked as highlighted)
- Line 43: How would we know about a rectal injury if it remains undetected altogether?
Respond 5: Thank you pointing this out. Postoperative diagnosis of rectal injury often presents significant diagnostic challenges. While peritonitis, fever, and abdominal pain or distension are the most frequently observed symptoms, other clinical manifestations such as ileus, gastrointestinal bleeding, fecal incontinence, passage of enteral contents through the skin, urethra, or vagina, or the presence of urine in the rectum may also occur. Additionally, systemic signs including tachycardia, hypotension, leukocytosis, and leukopenia.The paragraph was added to introduction section. Page number 2, line number 63-72 (marked as highlighted)
- Line 78: you have written that once the RI was confirmed, the RP procedure was completed. The RP procedure was not completed as the rectal repair was performed after specimen retrieval before vesico-urethral anastomosis.
Respond 6: Agree. The paragraph was revised according to reviwer’s suggestion. Page number 3, line number 115-119 (marked as highlighted)
- Line 91: some typo (surgical protocol.terile saline.)
Respond 7: Agree. We changed to .. sterile saline. Page number 3, line number 125 (marked as highlighted)
- Line 108: You have mentioned that none of the patients with a RI had undergone previous prostatic surgery or preoperative radiotherapy or hormonal therapy. How many patients in your series of 382 patients had this history?
Respond 8: Thank you pointing this out. Among our patient cohort, 10 had a history of TURP, and none had prior radiotherapy or hormone therapy. Page number 4, line number 163-166 (marked as highlighted)
- In table 1 (line 114): Spelling of complications is wrong.
Respond 9: Agree. The sentence was revised. Page number 4, line number 169 (marked as highlighted)
- Table 1 (line 114): Change Post-operative 8th day perform colostomy to Colostomy performed on the 8th post-operative day.
Respond 10: Agree. The sentence was changed according to reviwer’s suggestion. Page number 4, line number 169 (marked as highlighted)
- Line 118: Instead of general surgery surgeon, better to say gastro-intestinal surgeon or general surgeon.
Respond 11: Agree. The sentence was revised according to reviwer’s suggestion. Page number 5, line number 187 (marked as highlighted)
- Line 119: you have mentioned that four injuries were in the anterior region of the rectum; where were the other injuries?
Respond 12: The other rectal injury occured included two cases during prostatic apical dissection in four patients, one at the anterolateral rectal wall, one at the lateral rectal wall, and one involving the rectourethral muscles. This paragraph was added to results section. Page number 5, line number 189-190 (marked as highlighted)
- Line 122: change No patients were readmission due to complications to No patients needed readmission due to complications.
Respond 13: Thank you for pointing this out. No patients who experienced rectal injuries required hospital readmission within the 30-day postoperative period due to related RP complications. The sentence was revised. Page number 6, line number 193-194 (marked as highlighted)
- Line 125: what do you mean by mean pathological prostate volume?
Resond 14: Among patients with RIs, the mean weight of the surgical specimens (i.e., prostate and seminal vesicles) was 49.0 ± 10.3 mL. The sentence was revised. Page number 6, line number 196-197 (marked as highlighted)
- Line 128: LVI means lymphovascular invasion
Respond 15: Thank you for pointing this out. We mentioned lymph node metastasis (LNM) and abbreviation was changed. Page number 6, line number 199-202 (marked as highlighted)
- Line 136: you have said RI incidence was significantly higher in low-volume centres and in low-volume surgeons. I think you mean in low-volume centres performed by low-volume surgeons.
Respond 16: Agree. The sentence was changed according to reviewer’s suggestion. Page number 8, line number 249-267 (marked as highlighted)
- Line 196: change this sentence, To identify the extent of RI, using DRE further enhances. I suggest, DRE further helps to identify the extent of RI.
Respond 17: Agree. The sentence was changed according to reviewer’s suggestion. Page number 11, line number 378 (marked as highlighted)
- Your post-op suggestions in Discussion do not match what you have described in your care path way. Line 204 you have mentioned that clear liquid diet can be initiated on the first post- operative day but in the post-op care section (line 98) you suggest to generally start clear liquid diet on postoperative day 2. Please check. Why is cystogram needed in patients with rectal injury (Line 209)? Did you routinely do it? Is it a cystogram or a peri-catheterogram? In Discussion you suggest catheter for two weeks but you have kept for three weeks.
Respond 18: Thank you for this pointing out. We routinely keep the patients' catheters for 3 weeks. We prefer to removed catheter only after ensuring the integrity of the anastomosis radiographically. For this reason, we perform a cystogram and remove it if there is no leak. The 2-week catheter retention period in the discussion belongs to other studies. While conventional postoperative protocols recommend initiating oral liquid intake as early as postoperative day 1, our clinical practice adopts a more cautious approach for patients with RI, delaying clear liquid diets until postoperative day 2 to mitigate risks of early enteral stimulation on healing tissues.
The methods and discussion sections of the mentioned topic have been revised. Page number 11, line number 389-392 (marked as highlighted)
- In Abbreviations: Is DRE Digital rectal examination levels?
Respond 19: Agree. DRE means Digital rectal examination levels. Abbrevations was revised.
Comments on the Quality of English Language
Mentioned in the comments to Authors
Round 2
Reviewer 2 Report
Comments and Suggestions for Authors
The authors responded to all the queries sufficiently.
Author Response
Comment: The authors responded to all the queries sufficiently.
Respond: Thanks to reviewer for their valuable opinions and suggestions.
Reviewer 3 Report
Comments and Suggestions for Authors
The authors have presented a large data of rectal injury during radical prostatectomy and assessed the incidence, management, and long-term outcomes of this rare but serious complication at a single-center.
The corrections and changes are acceptable and good.
Few small suggestions:
- Line 63: You have said that RI can remain undetected altogether. I think you mean that RI can be missed intra-and peri-operatively and present late with infection or fistula.
- Line 97 & 170: What do you mean by digital rectal examination level. Is it level or DRE findings?
- What is International Index of Erectile Function short form 10? The standard IIEF has 15 questions and the abridged version is IIEF-5. What is short form 10? Even reference 11 mentions IIEF-5.
- Line 138: Instead of clear liquid diet the post operative day of surgery, say first post-op day.
- Line 171: the pathological Gleason score you have mentioned, is it pre-op biopsy or final histopathology?
- Line 189-190: there seems to be some error in this sentence. The other RIs included two cases during prostatic apical dissection in five patients (?), one at the anterolateral rectal wall, one at the lateral rectal wall, and one involving the rectourethral muscles
- Line 196: is ml a correct unit for weight?
- Spelling of intermediate is incorrect in Table 3
- Why do you need cystography in a patient with rectal injury?
Author Response
The authors have presented a large data of rectal injury during radical prostatectomy and assessed the incidence, management, and long-term outcomes of this rare but serious complication at a single-center.
The corrections and changes are acceptable and good.
Few small suggestions:
Comment 1: Line 63: You have said that RI can remain undetected altogether. I think you mean that RI can be missed intra-and peri-operatively and present late with infection or fistula.
Respond 1: We agree with this comment. We have revised the sentence according to reviewer’s suggestion. Mention exactly in the revised manuscript this change can be found – page number 2 and line 63-64 (marked as highlighted).
Comment 2: Line 97 & 170: What do you mean by digital rectal examination level. Is it level or DRE findings?
Respond 2: Thank you for pointing this out. In line 97 we mentioned about DRE levels and findings, and in line 170 we mentioned about only DRE findings. We have revised the manuscript – page number 3 and line 98 and page number 4 and line 173 (marked as highlighted).
Comment 3: What is International Index of Erectile Function short form 10? The standard IIEF has 15 questions and the abridged version is IIEF-5. What is short form 10? Even reference 11 mentions IIEF-5.
Respond 3: Thank you for ponitg with this out. In the manuscript, we referenced the International Index of Erectile Function short form 5 (IIEF-5). However, the International Index of Erectile Function short form 10 was inadvertently mentioned incorrectly. We have now corrected this error to ensure accuracy and consistency throughout the text. Page number 3 and line 106 (marked as highlighted).
Comment 4: Line 138: Instead of clear liquid diet the post operative day of surgery, say first post-op day.
Respond 4: We agree with this comment. We have revised the sentence according to reviewer’s suggestion. Mention exactly in the revised manuscript this change can be found – page number 3 and line 139 (marked as highlighted).
Respond Comment 5: Line 171: the pathological Gleason score you have mentioned, is it pre-op biopsy or final histopathology?
Comment 5: We agree with this comment. We have revised the sentence according to reviewer’s suggestion. Mention exactly in the revised manuscript this change can be found - page number 3 and line 99-100, page number 4 and line 174 (marked as highlighted).
Respond 6: Line 189-190: there seems to be some error in this sentence. The other RIs included two cases during prostatic apical dissection in five patients (?), one at the anterolateral rectal wall, one at the lateral rectal wall, and one involving the rectourethral muscles
Respond 6: We agree with this comment. We have revised the sentence. We agree with this comment and revised the sentence. Other rectal injuries comprised two cases occurring during apical prostate dissection, one involving the anterolateral rectal wall, one affecting the lateral rectal wall, and one involving the rectourethral muscles. Mention exactly in the revised manuscript this change can be found - page number 5 and line 190-192 (marked as highlighted).
Comment 7: Line 196: is ml a correct unit for weight?
Respond 7: We agree. We corrected this unit. Mention exactly in the revised manuscript this change can be found – page number 6 and line 199 (marked as highlighted).
Comment 8: Spelling of intermediate is incorrect in Table 3
Respond 8: We agree. We corrected this. Mention exactly in the revised manuscript this change can be found – page number 7 and line 211 (marked as highlighted).
Comment 9: Why do you need cystography in a patient with rectal injury?
Respond 9: Thank you for ponitg with this out. Therefore, cystography was routinely performed in patients with rectal injury to confirm or rule out the presence of rectourethral fistula. Mention exactly in the revised manuscript this change can be found – page number 4 and line 144-146 (marked as highlighted).